# Finding Differences Between Transformers and ConvNets Using Counterfactual Simulation Testing

**Nataniel Ruiz**
Boston University
`nruiz9@bu.edu`

**Sarah Adel Bargal**
Georgetown University
`sarah.bargal@georgetown.edu`

**Cihang Xie**
University of California
Santa Cruz
`cixie@ucsc.edu`

**Kate Saenko**
Boston University
MIT-IBM Watson AI Lab
`saenko@bu.edu`

**Stan Sclaroff**
Boston University
`sclaroff@bu.edu`

## Abstract

Modern deep neural networks tend to be evaluated on static test sets. One short-coming of this is the fact that these deep neural networks cannot be easily evaluated for robustness issues with respect to specific scene variations. For example, it is hard to study the robustness of these networks to variations of object scale, object pose, scene lighting and 3D occlusions. The main reason is that collecting real datasets with fine-grained naturalistic variations of sufficient scale can be extremely time-consuming and expensive. In this work, we present *Counterfactual Simulation Testing*, a counterfactual framework that allows us to study the robustness of neural networks with respect to some of these naturalistic variations by building realistic synthetic scenes that allow us to ask *counterfactual questions* to the models, ultimately providing answers to questions such as *"Would your classification still be correct if the object were viewed from the top?"* or *"Would your classification still be correct if the object were partially occluded by another object?"*. Our method allows for a fair comparison of the robustness of recently released, state-of-the-art Convolutional Neural Networks and Vision Transformers, with respect to these naturalistic variations. We find evidence that ConvNext is more robust to pose and scale variations than Swin, that ConvNext generalizes better to our simulated domain and that Swin handles partial occlusion better than ConvNext. We also find that robustness for all networks improves with network scale and with data scale and variety. We release the Naturalistic Variation Object Dataset (NVD), a large simulated dataset of 272k images of everyday objects with naturalistic variations such as object pose, scale, viewpoint, lighting and occlusions. Project page: `https://counterfactualsimulation.github.io`

## 1  Introduction

Testing computer vision models is a challenging endeavour. In order to claim the superiority of a model, the computer vision community usually studies validation accuracy on the ImageNet [12] dataset. Comparing models uniquely using Top-1 and Top-5 accuracy on this dataset can result in an incomplete evaluation of the advantages and disadvantages of each model. Recently, Vision Transformers (ViTs) [15] have been proposed as an alternative deep neural network model to rival Convolutional Neural Networks (ConvNets) [39] for computer vision tasks.

36th Conference on Neural Information Processing Systems (NeurIPS 2022).

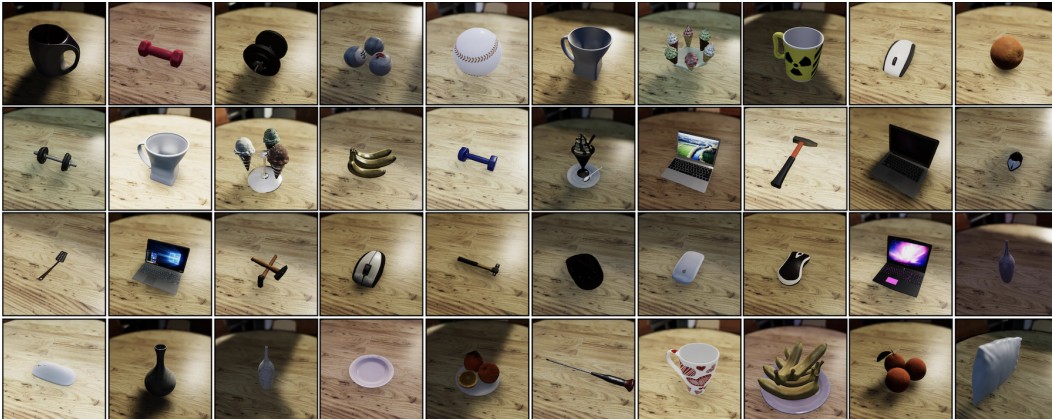

Figure 1: A sample of objects in our proposed **Naturalistic Variation Object Dataset (NVD)** in canonical pose with different lighting. NVD includes 272k images of object pose, scale, viewpoint and occlusion naturalistic variations of 92 object models with 27 HDRI skybox lighting environments.

ViTs have achieved impressive results, rivaling ConvNets and sometimes outperforming them in ImageNet accuracy. As it stands, if we restrict training data to ImageNet-22k, BEiT-L [5], a ViT variant, stands at the top of the ImageNet-1k leaderboard, with ConvNext-XL [42], a ConvNet, as a close second. Although the competition between these two classes of architectures has not yet been decided, there have been high-profile studies of potential advantages of ViTs compared to ConvNets. ViTs are believed to be more robust to certain adversarial attacks compared to ConvNets [58, 2, 8]. Although, recent work has also shown that this is not necessarily the case when ConvNets adopt the training recipes of ViTs [4]. Also, ViTs are believed to be more robust to domain generalization than CNNs [48, 4] and against occlusion, as described in Naseer et al. [46]. This last work shows that ViTs are more resistant than ConvNets to different types of patch removal from images, which is a way to simulate occlusions. One limitation of this study is that patch removal is a convenient but not fully realistic proxy for occlusion. It is natural to use such a proxy since datasets to study the effects of occlusion on object recognition and detection are scarce.

A more general limitation to works that compare properties of ViTs and ConvNets is that, even though they try to compare models of similar sizes and ImageNet accuracies, they do not account for the fact that *the compared ConvNets use slightly out-of-date design and training recipes.* In their impressive work, Liu et al. [42] propose a new class of ConvNets with modernized architecture design that seeks to closely resemble the design of ViTs, yet only uses convolutional layers. This work allows for a closer inspection of whether Transformers are superior to ConvNets due to the difference in inductive biases between transformer and convolutional layers.

We believe that studying the differences that arise in learned representations between Transformers and ConvNets to natural variations such as lighting, occlusions, object scale, object pose and others is important. A priori, convolutional and transformer layers have different inductive biases that should manifest themselves in different performance characteristics. For example, there have been conjectures that ViTs should outperform ConvNets with respect to partial occlusion since the transformer layers allow for early capture of long-range dependencies.

Until now, there have been two main obstacles that prevent the careful study of these differences: (1) Transformer and ConvNet architectures were not comparable in terms of overall design techniques and training recipe details, entangling these differences with transformer vs. convolutional layer differences (2) there is a scarcity of datasets that include fine-grained naturalistic variations of object scale, object pose, scene lighting and 3D occlusions, among others.

We propose an attempt to bridge these two obstacles, and strive to tackle the fundamental question:

> *Between Transformers and ConvNets; which of these models is more robust to naturalistic scene variations such as object pose, object scale, camera viewpoint, lighting and occlusions?*

In order to overcome (1) we compare the ConvNext [42] convolutional architecture to the Swin [41] Transformer architecture. Naturally, our contributions lie in our answers to obstacle (2). For this, we

propose a method to test a computer vision architecture in a counterfactual manner using simulated images. We call this method Counterfactual Simulation Testing. Specifically, our method allows us to ask *counterfactual questions* to the models, ultimately providing answers to questions such as *"Would your classification still be correct if the object were viewed from the top?"* or *"Would your classification still be correct if the object were partially occluded by another object?"*. This allows us to abstract from the base rate of domain gap between synthetic and real images for each architecture and to compare them fairly.

Using Counterfactual Simulation Testing we find evidence of performance differences between comparable ConvNets and Transformers with respect to object viewpoint, camera viewpoint, object scale and occlusions. We observe that consistently across different network sizes ConvNext is on average more robust than Swin with respect to *object pose* and *camera rotations*. We also observe that ConvNext architecture usually outperform Swin architectures in terms of recognizing small scale objects. Aditionally, we find that Swin and ConvNext architectures are roughly equivalent in terms of robustness with respect to occlusion, with Swin pulling ahead of ConvNext for severe occlusion. Finally, we find that the robustness of both architectures suffers greatly from naturalistic variations of the test data - and that robustness *improves* with network scale and with data scale and variety. In order to find these differences we generate five different realistic synthetic test sets of objects using the MIT ThreeDWorld (TDW) [19] platform with these specific naturalistic scene variations. The full dataset, named Naturalistic Variation Object Dataset (NVD), contains 272k images of 92 object models with 27 HDRI skybox lighting environments in an indoor scene.

In summary, our contributions include:

**Counterfactual Simulation Testing**, a method to test computer vision models for robustness with respect to naturalistic scene variations in a counterfactual manner using simulated images by varying scene parameters one-at-a-time and evaluating the stability of predictions.

**Naturalistic Variation Object Dataset (NVD)**, a dataset containing 272k images of 92 object models with 27 HDRI skybox lighting environments in a kitchen scene with 5 subsets of naturalistic scene variations: object pose, object scale, 360° panoramic camera rotation, top-to-frontal object view and occlusion with different objects. We hope that this dataset will allow for evaluation of modern computer vision models with respect to generalization to the synthetic domain and robustness to the naturalistic variations contained therein. A sample of NVD is shown in Fig. 1. We release this dataset to the public for use in benchmarking and architecture comparison.

## 2    Counterfactual Simulation Testing

**Testing robustness to natural variations in data.**    One major obstacle in evaluating models is that often, an aggregate metric, such as top-1 accuracy, will be used for comparison purposes. These metrics can give a certain sense of the power of the model, but can hide intricacies in performance with respect to natural data variations. For example, it is not possible to know to what degree a model is robust to occlusion given top-1 an top-5 ImageNet accuracy metrics.

We tackle this problem by generating large amounts of realistic synthetic data, due to the dearth of real datasets exploring these variations. An object recognition network can be written as $y = f(x)$, where $y$ is the predicted label from the image $x$ by the network $f$. The network $f$ can be either a convolutional neural network of a vision transformer. Our scene generator can be expressed as follows: $x = g(\theta^i, \psi^i, \kappa_0^i)$, where $g$ is the simulator, $\theta^i$ is the variable scene parameter of interest (e.g. object pose, scale, etc.), $\psi^i$ are the constant parameters controlling the main object model type, occluder object model type and lighting environment and $\kappa_0^i$ are the constant scene parameters (kitchen objects in scene, camera focal length, field of view, etc.). The variable $i$ denotes the different trials where the selection of variable scene parameter $\theta^i$ changes (thus $\kappa_0^i$ also changes).

In terms of scene content, in our work, only the main object (and occluder objects) vary in the scene. The rest of the scene is a pre-designed indoor scene. For same $\theta_i$ we generate different scenes with different object models (determined by $\psi^i$) and different lighting environments (also in $\psi^i$) for diversity purposes. For different scene variations encoded in $\theta_i$ we select object occlusions, object scale, object rotation around its vertical axis, camera elevation and camera panoramic rotation around the main object (i.e. $i = 5$). We describe the details of the dataset in Section 3.

Once we have generated such a dataset in this manner, we would like to test our network for robustness with respect to all selected variations $\theta_i$. One way is to test the network on all generated images and produce an expectation metric (averaged over $\psi^i$) with respect to the variable scene parameter $\theta_i$:

$$M_f(\theta^i) = \mathbb{E}_{\psi^i}[m[f(x_{\theta^i}), \hat{y}]], \tag{1}$$

where $x_{\theta^i} = g(\theta^i, \psi^i, \kappa_0^i)$, $\hat{y}$ is the true label, and $m$ can be any metric. Naturally, we could use top-1 or top-5 accuracy as this metric and we could plot $M$ with respect to $\theta^i$. This could allow us to compare different networks to find which network is more robust. Unfortunately, this solution *gives rise* to another important problem: domain generalization. By comparing two networks $f$ and $h$, we seek to understand their differences when applied to real data. Yet both models might have different generalization performance to the synthetic domain, having been trained on real data. This render a comparison of $M_f$ and $M_h$ unfeasible.

**How to address the Real to Synthetic domain gap.** Comparing pre-trained models out-of-the-box on a synthetic domain, even if highly realistic like ours, will elicit differing performances given the capabilities of models to generalize to this out-of-distribution domain. In order to avoid the brunt of this issue, we propose a way of asking counterfactual questions such as *"Is your answer still correct when I rotate the object by 60 degrees?"* or *"Is your answer still correct when I shrink the object to half of its size?"* to a model. As opposed to naively computing expected metrics, this abstracts from all prediction failures due to the model's inability to classify an object in the simulated domain under ideal circumstances due to the domain gap. This allows us to ask the deeper question *"On average, is ConvNext-Small better at recognizing small objects compared to Swin-Small?"*.

Let $\tilde{\theta}^i$ be the reference condition with respect to which we will ask all counterfactual questions. In most cases, this condition should be selected to be the average ideal condition for object recognition. For example, with respect to pose, this should be the canonical pose that elicits highest model accuracy. We compute the *proportion of correct conserved predictions* (PCCP) metric with respect to this reference condition as follows:

$$C_f(\theta^i) = \mathbb{E}_{\psi^i}\left[\frac{\sum_{\theta^i \in S^i} \mathbb{1}(f(x_{\tilde{\theta}^i}) = f(x_{\theta^i}))}{n(S^i)}\right], \tag{2}$$

where $S^i$ is the set of all $\theta^i$ in which $f(x_{\tilde{\theta}^i}) = y$, and $n(S^i)$ is the cardinality of $S^i$. We can then study this metric under variable $\theta^i$ and compare different models using point estimates, integrals over intervals (or spaces) of $\theta^i$, as well as plots of the metric. Under our counterfactual framework, computing $C_f(\theta^i)$ is effectively asking the question *"What proportion of your answers would still be correct if $\tilde{\theta}^i \to \theta^i$ happened?"*. In this way, PCCP metrics of different networks are comparable quantities of the relative robustness of the network's predictions with respect to the reference condition. In our problem we select reference conditions that present the easiest conditions for object recognition in order to have a large amount of initial correct predictions.

In summary, this approach, along with our selection of object models that are easy to classify, and high performance of our tested networks on these objects in the synthetic domain under canonical pose and ideal lighting, allows us to abstract from the synthetic domain gap. We can also consider different metrics such as *stability of predictions*, or even *proportion of incorrect predictions that become correct*. We discuss these in the supplementary material, we limit ourselves to PCCP in our main work since it suffices to study differences between ConvNets and Transformers.

## 3 Naturalistic Variation Object Dataset (NVD)

NVD is composed of five different subsets which seek to study naturalistic variations of scenes for object classification. Specifically, object pose, object scale, $360°$ panoramic camera rotation, top-to-frontal object view and occlusion with different objects. NVD is built using our customizable scene generator built on top of the MIT ThreeDWorld (TDW) [19] platform.

**Main objects.** We first study the intersection of the ImageNet-1k label space with the available object models in the TDW model library. After selecting all context appropriate object types we end up with 18 model classes. From these classes, we filter the 'iPod' and 'paintbrush' classes due to very low generalization accuracy across all architectures. The final classes are: *'banana'*,

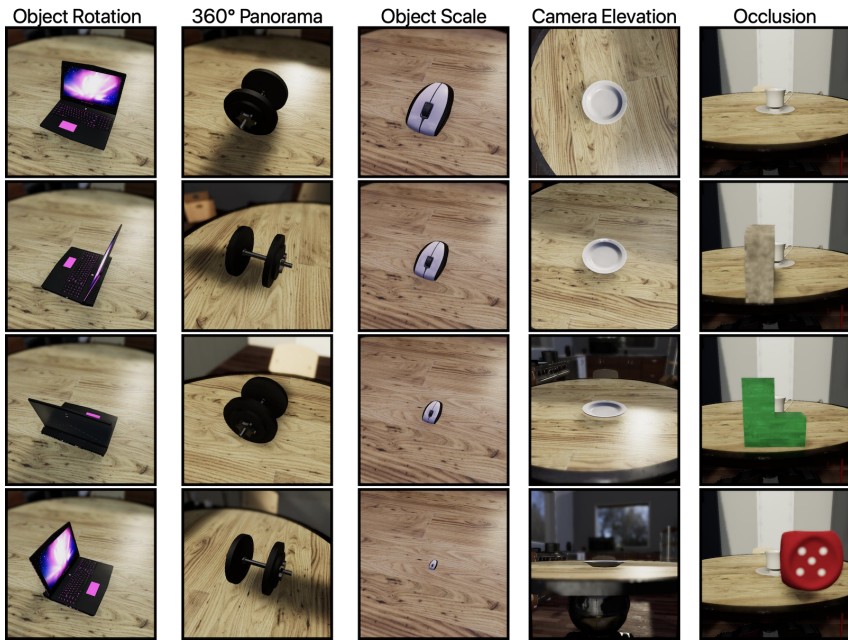

Figure 2: NVD includes the following naturalistic variations: *object rotation*, *360° panoramic camera rotation*, *object scale*, *top-to-frontal object view* and *occlusion* with different objects. All variations are performed on 92 object models under 27 different lighting environments.

'baseball', 'cowboy hat, ten-gallon hat', 'cup', 'dumbbell', 'frying pan, frypan, skillet', 'hammer', 'ice cream, icecream', 'laptop, laptop computer', 'microwave, microwave oven', 'mouse, computer mouse', 'orange', 'pillow', 'plate', 'screwdriver', 'spatula' and 'vase'. We found 99 object models in the TDW model library that are contained in this set of classes. We filter out 7 object models due to low quality or inconsistent size with other objects. In total we use 92 models to generate NVD. The TDW model library provides high-quality objects that are useful for studying object recognition due to the level of realism that is hard to find in open-source projects. A subset of the models can be seen in Fig. 1. Due to the relatively limited amount of realistic indoor object models in this library, some classes have more diversity in terms of different object models than others. We provide details on the amount of objects per class in the supplementary material.

**HDRI skybox lighting environments.** In order to increase the amount of variability we light the scene using 27 different HDRI skybox lighting environments that span the range of dark to oversaturated and contain unique features such as overhead artificial lighting, shadows, colored sunlight/moonlight, among others. We use the InteriorSceneLighting addon for TDW for this, and the 27 pre-defined available HDRI skyboxes from TDW. Some examples of different lighting can be observed in Fig. 1 and examples of all of them are included in the supplementary material.

**Naturalistic variations.** We generate 5 subsets of NVD with variations in object pose, panoramic camera rotation, object scale, top-to-frontal object view and occlusion with different objects. For object pose, we perform a full rotation of the main object around its vertical axis with step size of $15°$. For $360°$ panoramic camera rotation we perform a full camera rotation around the main object with step size of $30°$, for object scale we scale the main object by $s = u(o) \times 0.25$, where $u(o)$ is the unit scale of object $o$ computed using the TDW *get_unit_scale* function. We multiply by $0.25$ in order for the objects to be scaled in line with the context objects in the kitchen scene. Then we vary this scale multiplier by multipying it by a scale factor in the $[0.2, 1]$ interval, with step size $0.05$. For top-to-frontal object view, we vary the camera elevation in the $[0°, 90°]$ range with step size of 5. For occlusion, we position an occluder object between the camera and the main object, and vary its x-axis position, such that it starts at the left of the frame and ends at the right, passing directly between the camera center and the main object. Thus, the main object is occluded by the occluder in a variable manner. We use three different occluder objects that have very distinct visual characteristics and occlude the object in different manners: a stone bookend, a red die and a green

Table 1: Architectures used in our paper, with model details and corresponding mean accuracies on the ImageNet-1 validation set (IN) and the entirety of NVD.

| Architecture | #param. | FLOPs | IN top-1 | NVD top-1 | NVD top-5 |
|---|---|---|---|---|---|
| ConvNext-T | 28M | 4.5G | 82.1 | 24.7 | 46.7 |
| Swin-T | 28M | 4.5G | 81.2 | 21.1 | 41.1 |
| ConvNext-S | 50M | 8.7G | 83.1 | 27.7 | 50.0 |
| Swin-S | 50M | 8.7G | 83.2 | 22.1 | 42.9 |
| ConvNext-B | 89M | 15.4G | 83.8 | 23.5 | 50.2 |
| Swin-B | 88M | 15.4G | 83.5 | 22.7 | 45.2 |
| ConvNext-B-22k | 89M | 15.4G | 85.8 | 34.0 | 58.9 |
| Swin-B-22k | 88M | 15.4G | 85.2 | 31.5 | 56.6 |
| ConvNext-L-22k | 198M | 34.4G | 86.6 | 43.1 | 67.6 |
| Swin-L-22k | 197M | 34.5G | 86.3 | 36.3 | 61.4 |

L-shaped block. These objects do not have a corresponding label in the ImageNet-1k dataset, and thus do not act as first-order distractors with respect to the main object, although they may have second-order associations with classes in ImageNet and thus may still act as distractors.

## 4 Experiments

For all experiments, we compare ConvNext architectures with their "twin" Swin architectures. ConvNext and Swin network both have five overlapping architectures that have extremely similar number of parameters and ImageNet validation accuracy. For a more detailed inspection of these statistics, see Table 1. We use the official open-sourced code for both models, as well as the public model weights. All PCCP metrics in this section are computed using top-5 predictions for greater stability of results. We plot standard deviation error bars for the counterfactual study figures using bootstrap resampling (100 resamples). We use two GeForce RTX 2080 GPU to perform all experiments.

**ConvNext networks generalize better to the NVD synthetic domain than Swin Transformers.** We compare the mean performance of twin ConvNext and Swin architectures in Table 1. Firstly, we observe that all twin architectures have almost identical model sizes and FLOPs. They also have very comparable ImageNet accuracies, with some architecture pairs only differing by tenths of a percentage point. We observe that all ConvNext models obtain higher mean accuracy on NVD than their twin Swin models. In particular, even when the corresponding ConvNext network performs worse than the twin Swin network on ImageNet (e.g. ConvNext-S/Swin-S) the ConvNext networks performs better on NVD. This is a surprising finding that goes against the belief that Transformers are more robust to OOD generalization with findings of greater OOD generalization detailed in Bai et al. [4] and Paul et al. [48]. The improvement in OOD generalization in ConvNext networks is most likely a consequence of the modernized design of the architecture, since Bai et al. [4] align training recipes in their work.

**Performance of all networks sharply decreases with naturalistic variations.** Looking across all Figures 4,5,6,7,8 - we observe that in general naturalistic variations severely affect the performance of all networks. In the supplementary material we include an experiment on object rotation where networks are *finetuned* on the simulated objects and find similar results. This highlights that even though these networks achieve high accuracies in datasets like the ImageNet validation set, there are larger questions about the robustness of the network on other datasets as well as the robustness of the networks with respect to commonly encountered variations in the real world.

**For both ConvNext and Swin Transformers, network scale and data scale and variety improve robustness with respect to all the studied naturalistic variations.** Again, looking across all Figures 4,5,6,7,8 - we can see that on average robustness to naturalistic variations increases with network scale (e.g. Tiny network compared to a Base network) and data size and variety (e.g. networks trained on ImageNet-1k and networks pre-trained on ImageNet-22k). This gives a potentially simple path towards improving overall robustness: train larger networks with larger and more varied data.

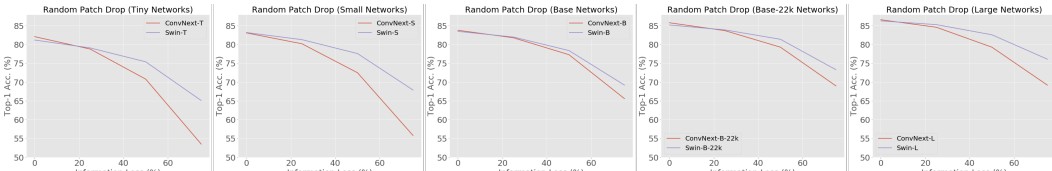

Figure 3: Random patch drop occlusion study on ConvNext and Swin networks on the ImageNet-1k validation set. Swin Transformers are slightly more robust to this type of artificial occlusion than ConvNext networks when the information loss is small, although they become comparatively much stronger as the information loss is increased.

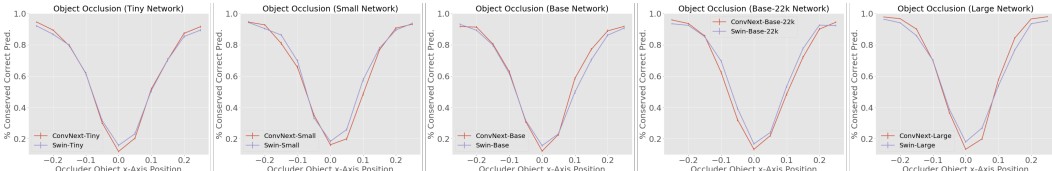

Figure 4: Counterfactual study of all sizes of ConvNext and Swin networks for occlusion of main object using occluder objects.

## 4.1 Counterfactual Simulation Testing

**Swin Transformers are more robust to occlusions than ConvNext networks.** Several works have claimed superiority of Transformers over ConvNets in the case of occlusions. Naseer et al. [46] compare ResNets to DeiTs by dropping random test image patches. They show that DeiTs are much more robust than ResNets to this transformation. Although DeiTs are superior to ResNets in this scenario, we can question whether (1) this is a general phenomenon brought about by the differences between conv layers and transformer layers and (2) this type of patch dropping actually approximates naturalistic occlusions due to other objects in a scene.

For (1), we provide evidence that this difference between ResNets and DeiTs is not due to the nature of transformer vs. conv layers. In order to do so, we re-run the random patch drop experiments in [46] on all pairs of ConvNext and Swin networks. We present results for different levels of information loss in Fig. 3. For all networks we observe a drop in performance as information loss increases. We show that ConvNext networks *do not* suffer the same critical failure mode that DenseNet121, ResNet50, SqueezeNet and VGG19 exhibit in Naseer et al. [46]. The top accuracy of these four classic ConvNets collapses to half after about 10% of patches are occluded and to 0 after about 40% information loss. ConvNext shows that it is much more resistant, conserving a large part of its accuracy even after 50% information loss. Next, we observe that ConvNext networks are slightly less robust than Swin Transformers for low amounts of information loss (under 50%). Thereafter, Swin becomes comparatively much more robust as occlusion becomes severe.

For (2), we conduct counterfactual simulation testing of ConvNext and Swin networks with variable occlusion. We generate images of the main object on the center table of the scene, and add an occluder object between the camera and the main object. The details of this subset are explained in Section 3. In this counterfactual experiment we compare all occluded scenes with the non-occluded initial scene where the occluder is not in the scene. We observe in Fig. 4 that both Swin and ConvNext tend to have similar failure responses to the variable occlusions and that performance drops precipitously, achieving a minimum at zero in the x-axis. It is important to note that for maximal occlusions (i.e. occluder object at zero in x-axis) all Swin networks exhibit stronger robustness than ConvNext networks. For many of the lesser occlusions, Swin networks are still slightly ahead of ConvNext networks. We conclude that Swin Transformers are slightly more robust to occlusions than ConvNext networks and this echoes the findings of our experiment on real ImageNet images with patch drop.

**ConvNext networks are more robust to object scale than Swin Transformers.** In order to compare the robustness of ConvNext and Swin networks to scale, we first generate images of all main object models with unit object scale. Although objects have different volume in this state (e.g. microwaves are larger than spatulas), they take a fair amount of space in the frame and don't present much of a challenge to large architectures. Specifically, *top-5 accuracies* for Swin-L and ConvNext-L

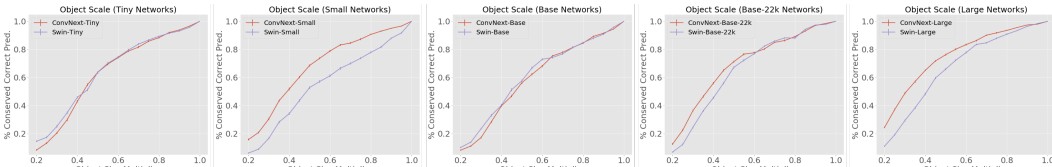

Figure 5: Counterfactual study of all sizes of ConvNext and Swin networks for object scale.

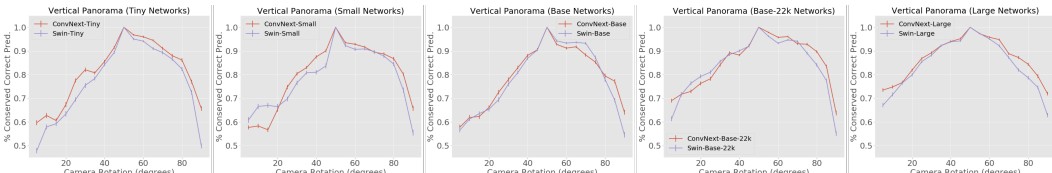

Figure 6: Counterfactual study of all sizes of ConvNext and Swin networks for frontal-to-top camera rotation around the main object.

are **90.5%** and **93.8%** respectively. We then perform counterfactual simulation testing by plotting the stability of correct predictions when the object scale multiplier is reduced from 1. We show these results in Fig. 5. We observe that for Small, Base-22k and Large-22k networks, ConvNext vastly outperforms Swin for smaller objects. For Base-22k and Large, the difference becomes large after the scale multiplier is under 0.6. For Tiny and Base networks, ConvNext and Swin perform in very similar manner, with Swin slightly outperforming ConvNext on very small objects by a small margin.

**ConvNext networks are more robust to rare top and frontal object views than Swin networks.** Here we compare the robustness of ConvNext and Swin networks to different viewpoints of the main object. Specifically, we start with a camera viewpoint that captures the main object from a fully-frontal view to a fully-top view. Object recognition is known to fail when the object is viewed from rare poses such as from the top [1]. In this experiment we compare all views to the canonical view with a camera pointing at the object with a 45° elevation.

We show in Figure 6, that on ImageNet-1k and ImageNet-22k-trained ConvNext and Swin networks, this is still the case. Both network classes exhibit drastic drops in proportion of conserved correct predictions when the pose is modified from a canonical pose with a camera at 45° of elevation to either a fully-frontal pose at 0° or a fully-top view at 90°. We observe that both increasing network size and pre-training on ImageNet-22k increase robustness on both architectures, with the best model being ConvNext-L-22k which achieves a conservation of correct answers in top view of around 73%.

We observe as well that all ConvNext networks obtain a higher proportion of conserved correct predictions for views from the top (e.g. with high elevation) than comparable Swin networks. We conclude that ConvNext networks are more robust to rare top views of objects than Swin Transformers given comparable architecture sizes and identical training data and recipes. Finally, we find that most ConvNext networks have higher proportion of conserved correct predictions across different elevation angles, with some notable exceptions such as a plummeting robustness of ConvNext-S in low elevation, and slightly lower robustness of ConvNext-B and ConvNext-B-22k compared to their twin Swin networks in certain elevation intervals.

**ConvNext networks are, on average, more robust than Swin networks to different lateral object viewpoints.** In order to test the robustness of both classes of networks to different lateral views of objects, we conduct two counterfactual studies. The first scene variation consists in a panoramic 360° camera rotation around the object. In this scenario the lighting on the object can change drastically. Many lighting skyboxes in NVD contain a light source coming through a window behind the object, and when the camera is rotated we can either have a high contrast view of the object with the main light source behind the camera, or a shaded view of the object against the light. The second variation consists of a fixed camera setting with the main object performing a full 360° rotation around its vertical axis. This allows us to keep lighting fixed while we evaluate the networks on different poses for the same object. Using both experiments we can disentangle decreases in performance due to object pose or the interplay between object pose and lighting.

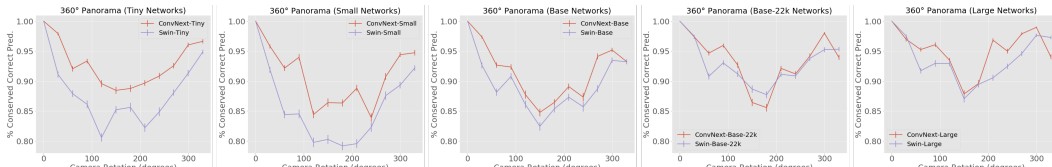

Figure 7: Counterfactual study of all sizes of ConvNext and Swin networks for panoramic 360° camera rotation.

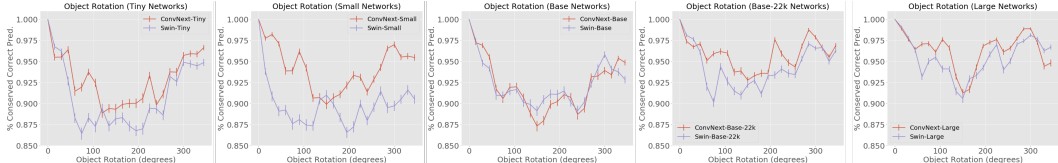

Figure 8: Counterfactual study of all sizes of ConvNext and Swin networks for 360° object rotation.

We plot the proportion of conserved correct predictions for the panoramic camera rotation in Fig. 7. First, we observe that both networks struggle to a higher extent with images taken on opposite to the position, with the light source behind the camera. This is due to two factors: the rear-view object are, in general, rare cases (e.g. the back of a microwave) and some lighting environments provide very strong light resulting in overexposed scenes with less object detail. Interestingly, we observe that for all networks ConvNext is more robust than Swin, with the slight exception of the 150° to 180° range for Base-22k networks. We also see in this experiment that curves are much less smooth and irregular. This is partly due to the nature of object rotations: some objects are easy to recognize in multiple different views, while others present very challenging views within small rotation intervals.

This experiment entangles lighting and object pose. In order to partially disentangle them, we perform a study where the camera is fixed and the main object is rotated along its vertical axis. We can observe in Fig. 8, that, on average, ConvNext is more robust to these rotations than Swin. Even so, the difference between both networks is lower than in the previous experiment, suggesting that ConvNext is more robust to different lighting and that this contributes to performance differences in Fig. 7.

## 5   Related Work and Discussion

**Simulation**   In recent years, the community has explored using synthetic data for training deep networks [18, 55, 54, 16, 50, 6]. Some work goes a step further and learns the distribution of data needed to improve model performance [56, 43, 20, 7, 32, 3]. There is an easy way of measuring success of these models by evaluating them on real data. There are also attempts to benchmark and test models using synthetic data [51, 45, 30, 6, 34, 57]. The synthetic-real domain gap complicates this endeavour, making it hard to generalize insights to the real-world. In contrast, we propose a counterfactual study of model failure using a photorealistic simulator. This allows us to abstract from the model domain gap base rate. A different direction of work that could allow for this is the domain adaptation literature which adapts pixels or features to bridge the gap [21, 9, 63, 62, 28, 49].

**ConvNet and Transformer comparative studies**   ConvNets have dominated the field of computer vision for the last decade [39, 37, 59, 25, 29, 60, 42]. Vision Transformers have recently proven to be extremely competitive, sometimes outperforming ConvNets in computer vision tasks [15, 61, 64, 65, 41, 5, 14]. There have recently been studies into the differences between ConvNets and Transformers. Among other studies [46, 67], Raghu et al. [52] study the internal representation of ViTs, finding early aggregation of global information and strong propagation of features from lower to higher layers. Bai et al. [4] show that CNNs can be as adversarially robust as ViTs, but lag behind in OOD generalization. In contrast to this work, we compare recent CNNs and ViTs with respect to naturalistic scene variations such as occlusion and object pose in a fine-grained manner, with some surprising findings. To the best of our knowledge, ours is the first work that seeks to compare these classes of architectures in this way and that proposes a way to equalize the (synthetic domain) playing field using counterfactual analysis.

**Robustness and counterfactual fairness** There is research on OOD robustness of networks including synthetic images [50], stylized images [22], corrupted images [26] and natural adversarial images [27]. Our effort complements this direction with a study of robustness with respect to natural scene variations. There exist investigations into network fragility regarding rare object poses [1] and occlusion [68, 36] - and work that tries to address the weaknesses [35]. Our work represents a serious attempt at providing a flexible and general method with which to study these types of variations, along with many others that are less common in the literature such as realistic lighting environment changes. Our counterfactual framework allows us to compare the robustness of different architectures that might not have the same base-level performance in a fair setting. There exists work on counterfactual fairness [38, 10, 31], a field that studies ML decision fairness from a counterfactual viewpoint. The closest literature to our work is the sparse literature of counterfactual interpretability [24] and sensitivity [11, 13, 57]. To the best of our knowledge, we are the first to propose a method to study object recognition networks for robustness with respect to natural scene variations by combining two key ingredients: a counterfactual approach and realistic synthetic data with fine-grained variations.

**Causal analysis** There exists recent work on causal analysis of learned representations by adding simple context-aware synthetic objects to real scenes [53]. The goal of the work is to study causal disentanglement in the latent spaces of VAE-based methods. Instead we directly study prediction robustness of modern neural networks with respect to naturalistic variations using a counterfactual approach. Our proposed NVD dataset differs from the CANDLE dataset, in that we simulate the entirety of the scene using realistic object models and have control over a larger amount of variation such as lighting and scene content. Other recent work [66] studies causality using time-consecutive images. The primary differences with our work are that we manipulate specific scene variations and can abstract from the time dimension. Finally, there are several works that explore synthetic generation of datasets with disentangled factors [33, 44, 17, 23, 40, 47]. This body of work sets out to study representation disentanglement with simple synthetic scenes, which we do not seek to address in our work.

# 6 Conclusion

We conduct a counterfactual comparative study of Swin Transformers and ConvNext networks by proposing NVD; a novel realistic synthetic dataset of naturalistic scene variations. We find that (1) ConvNext networks are more robust to the simulated domain shift than Swin transformers (2) ConvNext networks are more robust to scale and pose variations than Swin transformers (3) Swin transformers are more robust than ConvNext networks with respect to partial occlusion. We also find that robustness for all factors increases when network size increases (for both classes of networks) and when dataset size increases. We release NVD and our flexible and customizable scene generator.

**Limitations** Due to limited real data, we test networks on simulated data. It can be hard to generalize insight into the real world. We tackle this limitation by (1) generating a photorealistic dataset using state-of-the-art research software, with networks achieving very high accuracy on object recognition under ideal scene parameters (2) proposing counterfactual studies in order to further abstract from domain gap. General statements about ConvNets and Transformers are hard to make, given that architectures evolve. In contrast to previous explorations, we propose to study the most comparable architectures to date; ConvNext and Swin, due to their similar design choices and aligned training recipes. To further increase robustness, we study 4 different sizes of these networks with 2 different training datasets (ImageNet-1k and ImageNet-22k).

**Broader Impact** Our work can be used to address model bias, e.g. face analysis networks with a respective simulator. Interactions exist with applications that can negative for society, for example, knowing weaknesses of networks can lead to simple ways of attacking them using natural inputs. As always, improving computer vision models goes hand in hand with enabling applications such as malicious surveillance. We include a more detailed discussion in the supplementary material.

**Acknowledgments** We deeply thank Jeremy Schwartz and Seth Alter for their advice and support on the ThreeDWorld (TDW) simulation platform. This work was supported in part by NSF and DARPA grants to Kate Saenko and a gift grant from Open Philanthropy to Cihang Xie.

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
