# Supplementary Material: Finding Differences Between Transformers and ConvNets Using Counterfactual Simulation Testing

**Nataniel Ruiz**
Boston University
nruiz9@bu.edu

**Sarah Adel Bargal**
Georgetown University
sarah.bargal@georgetown.edu

**Cihang Xie**
University of California
Santa Cruz
cixie@ucsc.edu

**Kate Saenko**
Boston University
MIT-IBM Watson AI Lab
saenko@bu.edu

**Stan Sclaroff**
Boston University
sclaroff@bu.edu

## Counterfactual Metrics

In the main paper we study the *proportion of correct conserved predictions* (PCCP) metric with respect to a reference condition $\tilde{\theta}^i$ as follows:

$$C_f(\theta^i) = \mathbb{E}_{\psi^i}\Big[\frac{\sum_{\theta^i \in S^i} \mathbb{1}(f(x_{\tilde{\theta}^i}) = f(x_{\theta^i}))}{n(S^i)}\Big] \tag{1}$$

where $S^i$ is the set of all $\theta^i$ in which $f(x_{\tilde{\theta}^i}) = y$, and $n(S^i)$ is the cardinality of $S^i$. Although this metric suffices to study the degradation of correct predictions as the scenario becomes more challenging, there are other counterfactual metrics that we can consider that help in analysing characteristics of a prediction system. Specifically, we can compute a *proportion of all conserved predictions* (PACP) metric with respect to the reference condition:

$$C_f^a(\theta^i) = \mathbb{E}_{\psi^i}\Big[\frac{\sum_{\theta^i \in \Theta^i} \mathbb{1}(f(x_{\tilde{\theta}^i}) = f(x_{\theta^i}))}{n(\Theta^i)}\Big] \tag{2}$$

where $\Theta^i$ is the set of all $\theta^i$. This metric includes all incorrect predictions that remain incorrect.

**Swin is less affected by proximal context than ConvNext** Paradoxically, due to the capture of context by deep learning systems, adding an object in a scene as a partial occluder for example, can turn an incorrect prediction into a correct prediction. We plot PACP for all ConvNext and Swin network size pairs for the occlusion variation in Fig. 1. We observe an even stronger tendency for Swin to conserve initial predictions under partial occlusion. This leads us to an interesting finding, that ConvNext is not only slightly worse at handling occlusion, but that *any* prediction it makes is much more unstable with respect to occluding objects - even in the positive direction. This phenomenon suggests that *proximal context* weighs more into ConvNext's predictions, since some incorrect predictions of the object on the table become correct once another object (red die, stone bookend or green L-shaped cube) become visible in the scene.

**Finetuning Experiment** We finetune all networks on the simulated object classes in the canonical pose with bright lighting for 30 epochs at same learning rates (5e-5 for Tiny and Small networks, 2e-5 for Base and Large). We then run the same counterfactual study of 360 degree object rotation as in Figure 8 of the main paper. We show our experiment in Figure 2. We find very similar conclusions,

36th Conference on Neural Information Processing Systems (NeurIPS 2022).

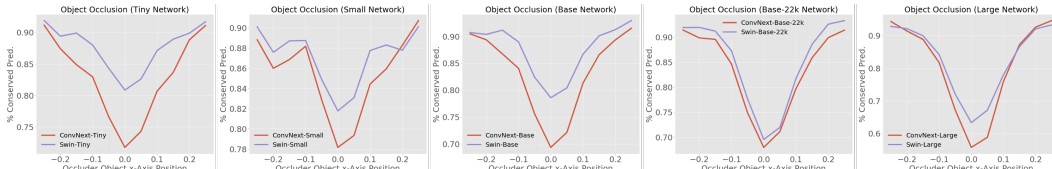

Figure 1: Proportion of all conserved predictions of all sizes of ConvNext and Swin networks for occlusion of main object using occluder objects.

with ConvNext networks outperforming Swin networks in robustness to different poses. One small difference is a collapse in performance for some small ConvNext networks for the rare pose of 180 degrees, where the object is fully turned around. This might be due to more aggressive overfitting of ConvNext to object features in the canonical pose.

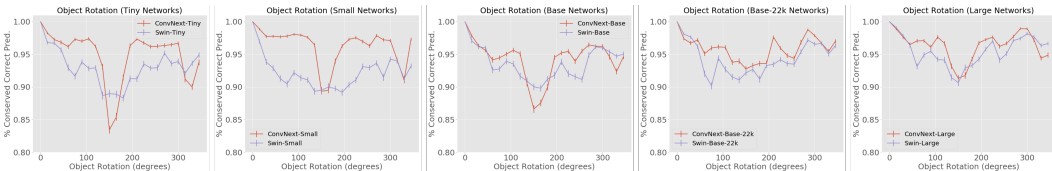

Figure 2: Counterfactual study of all sizes of ConvNext and Swin networks for 360 degree object rotation. Networks were finetuned on the simulated objects in a canonical viewpoint under a single lighting environment.

## NVD Dataset Details

Here we present more details about the proposed NVD dataset. In Fig. 3 we show a single object under all 27 available lighting environments in the NVD dataset. We can observe that lighting ranges from very bright, to dim, with many variations in shading, light direction, light intensity and light color. These variations are reflected in the appearance of the object, adding diversity to our dataset as well as complexity and another possible variation with which we can measure domain generalization. We highlight that this type of lighting variation is incredibly hard to achieve in a real world dataset, since the scene remains constant while only the lighting changes.

Next, in Fig. 4, we present a non-exhaustive showcase of the 92 object models contained in NVD. The full list of classes, with respective object model quantities is: *'banana' : 7*, *'baseball' : 3*, *'cowboy hat, ten-gallon hat' : 1*, *'cup' : 7*, *'dumbbell' : 9*, *'frying pan, frypan, skillet' : 3*, *'hammer' : 8*, *'ice cream, icecream' : 6*, *'laptop, laptop computer' : 4*, *'microwave, microwave oven' : 1*, *'mouse, computer mouse' : 10*, *'orange' : 4*, *'pillow' : 15*, *'plate' : 3*, *'screwdriver' : 3*, *'spatula' : 3* and *'vase' : 5*.

## Patch Drop Experiments

In Fig. 5 provide a visualization of different levels of information loss for the random patch drop experiment in Section 4.1 of the main paper. We observe an uncanny ability of Swin Transformers to correctly predict the classes of images with very high (75%) information loss, even when they become hard to recognize to humans. ConvNext performance collapses much faster, and this is consistent with our observation that Swin is less affected by proximal context.

## Swin Transformer V2 Experiments

The recently released Swin Transformer V2 architecture [5] obtains higher accuracies on ImageNet than the original Swin. Its main differences with the original Swin architecture are threefold: using normalization layers after attention layers, instead of before; replacing dot product attention with a scaled cosine formulation of attention; and changing the linear-spaced coordinates to log-spaced

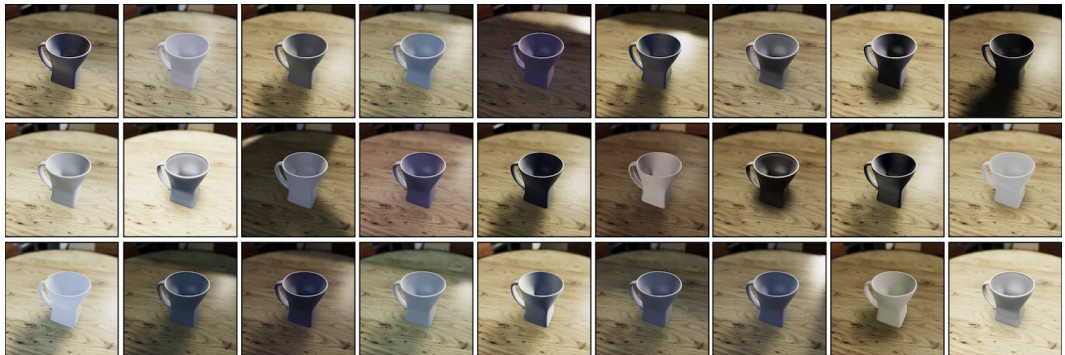

Figure 3: A showcase of all available lighting environments in the NVD dataset, ranging from very bright and low-contrast lighting to detailed shadows and different colored sunlight.

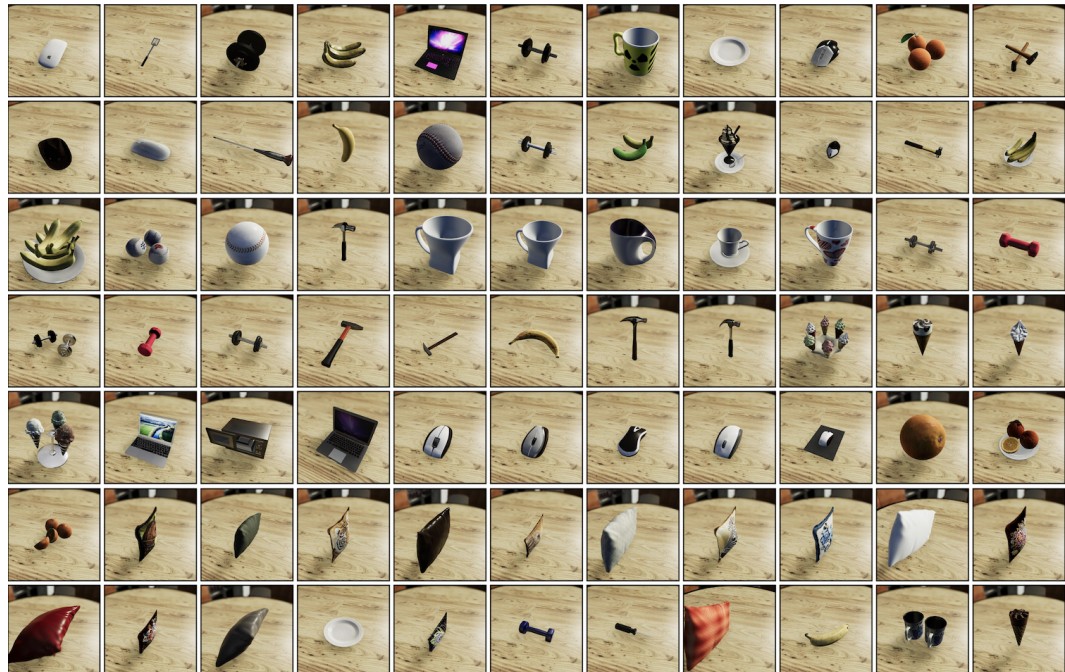

Figure 4: A non-exhaustive showcase of NVD objects under a constant lighting environment.

coordinates. These changes do not affect the number of parameters in the architecture, which would a priori allow for a fair comparison with ConvNext. Unfortunately, Swin V2 architectures are exclusively available for inference on images of size at least 256x256. This gives the architecture a slight unfair advantage over ConvNext and increases the FLOPs of each Swin V2 architecture far above same size ConvNext architectures. Nevertheless, for completeness, we conduct all of the studies in the main paper on Swin V2.

We compare the performances of Swin and Swin V2 for the occlusion, object scale, camera elevation, object rotation and 360° panorama variations in Fig. 6. We observe similar responses for both models across different naturalistic variations. For example, both models perform near identically for occlusions. We can see some outlier model sizes that present higher differences, such as the Small architectures which present near-identical response to occlusion but very different responses to all other variations - with Swin-V2-S being much more robust than Swin-S. One important thing to note is that going from 224x224 images to 256x256 images should be very advantageous for robustness with respect to object scale in particular - and we see this difference manifest itself for several sizes of Swin.

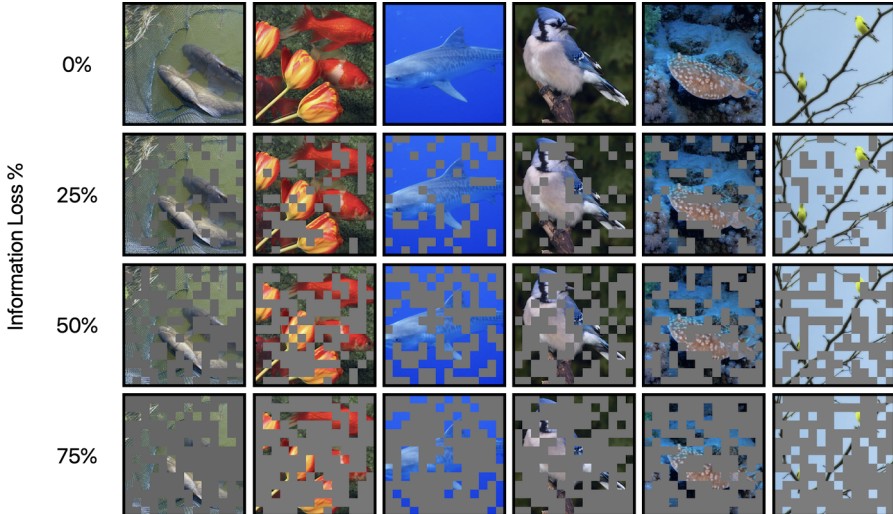

Figure 5: Illustration of random patch drop on ImageNet images with different percentages of information loss.

Finally, we present comparisons for all variations between ConvNext and Swin V2 in Fig. 7. Again, these comparisons are not fully fair given Swin V2's larger input size. We can verify some of our previous observations: even with a larger input size and architectural improvements Swin V2 is **less robust to occlusion** than ConvNext. For the object scale experiment, we see an improvement in performance from Swin V2, equalizing ConvNext across some comparisons. Again, an increase in input size can bias this comparison towards Swin V2. Across other tests we can see improvements from Swin V2, with more robustness to rare poses, although frequently outperformed by ConvNext. Again, Swin V2 Small seems to improve the most across tests.

**Studies With Top1 Accuracy**    We believe that top5 accuracy is a stronger metric for study than top1 in our setting, since NVD images contain first order distractors that are in the ImageNet label space. The primary distractor is the dining table where the objects are set. We decided to include these naturalistic distractors, in order to make the scene more realistic. An essentially blank scene without other objects would not make a convincing experiment. This means that analyzing top1 metrics is very noisy given that a network could output one of the distractors as its top1 prediction and this would not represent the overall power of the network which could be predicting both a distractor and the main object with high confidence. Nevertheless, here we include Figure 8 that uses top1 accuracy for our counterfactual experiments. When analyzing these experiments we observe network outputs and find that, even when a network predicts the correct main object with a high probability, sometimes the top1 answer is incorrect given that it also predicts the class "dining table, board" with higher confidence. Thus, we cannot give an analysis of these figures that confidently gives us a conclusion on performance differences.

## Broader Impact Extended Discussion

Here we include more discussion about the potential broader implications, both positive and negative of our work. For negative societal consequences, our work is entangled with all work that tries to improve discriminative computer vision systems. There are many possible malicious uses for such technology, ranging from surveillance and monitoring, to offensive military uses. Another particularly pernicious consequence of developing causal diagnostic tools like ours, is that good causal knowledge of a network can allow for easy naturalistic attacks on that network. For example, knowing our observation that the mere presence of a red die in a scene affects the predictions of ConvNext, can allow attackers to use small innocuous objects to bias the prediction of such a system. Such attacks can be performed in high-risk situations such as against autonomous navigation systems of vehicles in traffic scenes.

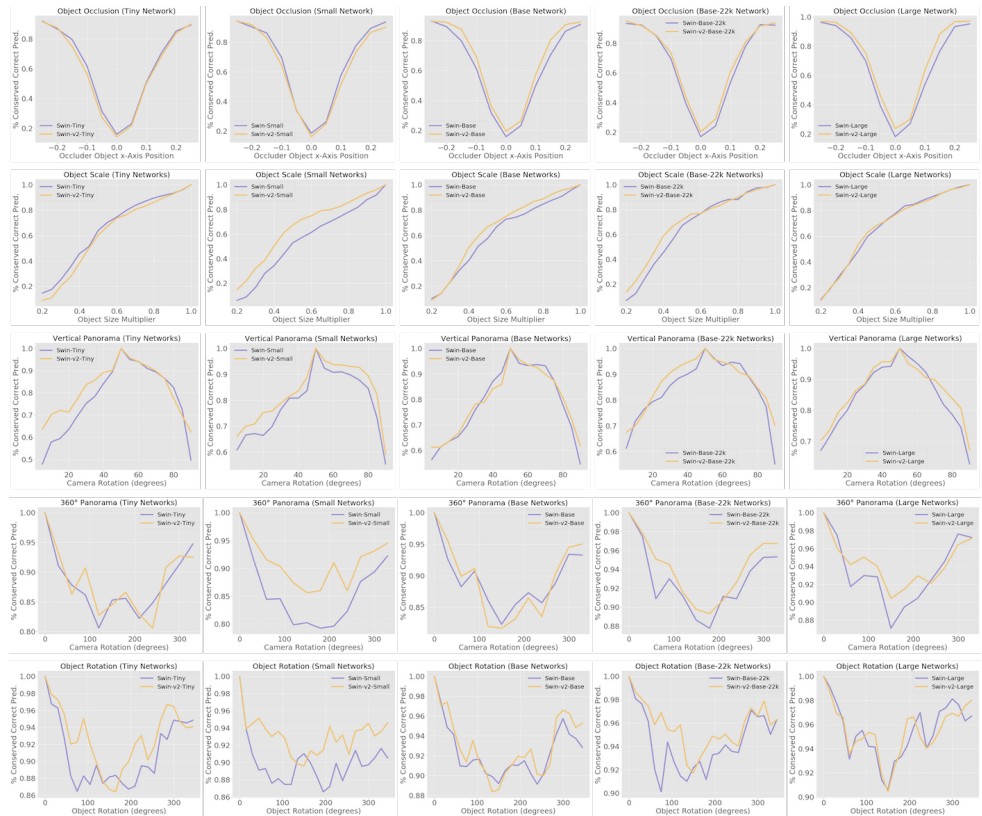

Figure 6: Swin and Swin V2 comparisons for all sizes on five different naturalistic scene variations.

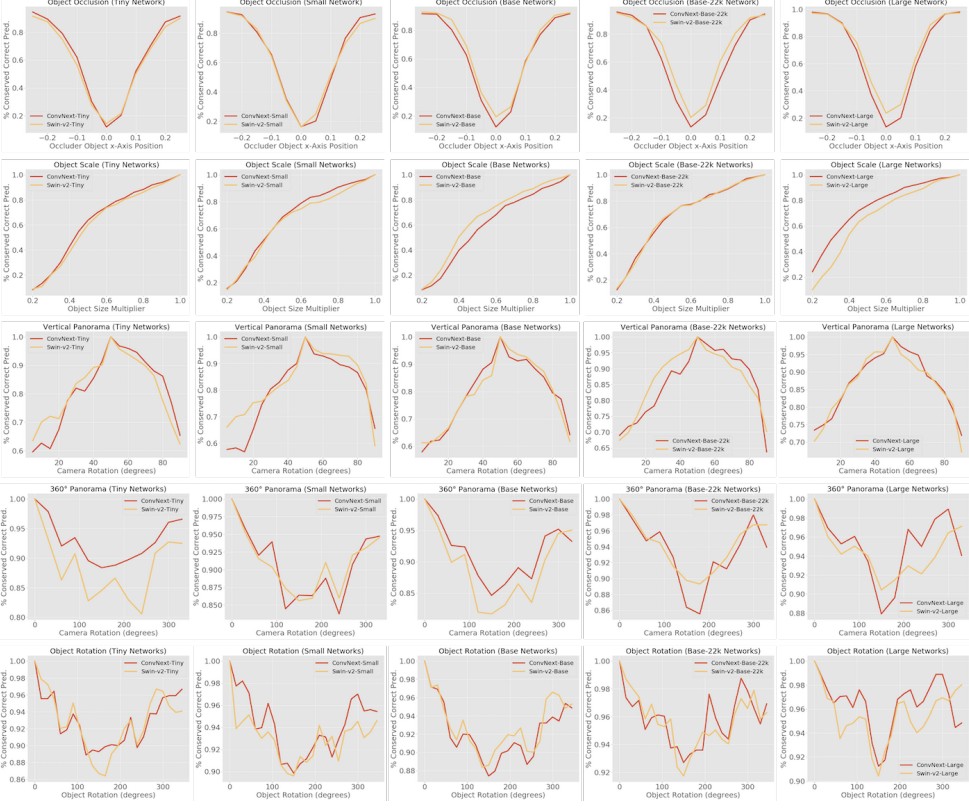

Figure 7: Counterfactual studies comparing ConvNext to Swin-V2 networks for all NVD variations.

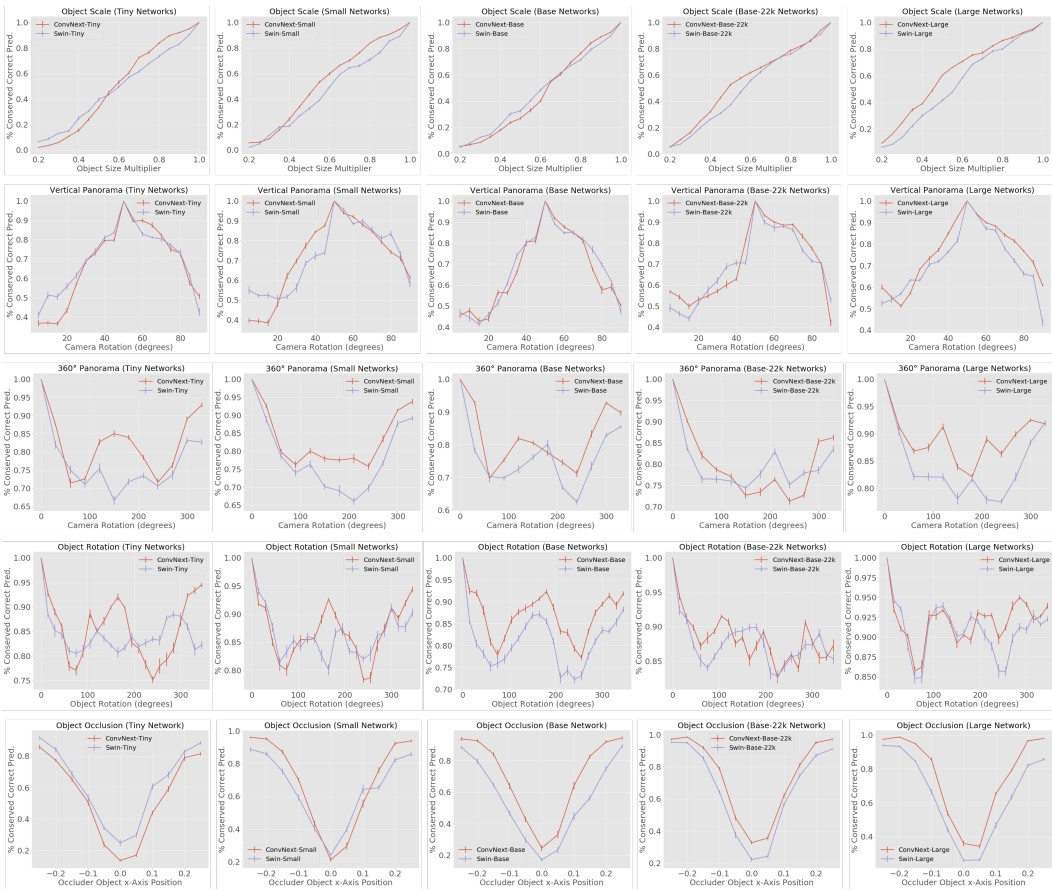

Figure 8: Counterfactual study for all variations using top1 accuracy. We note that top1 accuracy is an unreliable metric for comparison of networks in our setting given the existence of strong first-order distractors in the scene such as the dining table that supports the main objects. Therefore, this study may contain more noise than signal.

Our work also has a high number of positive societal applications. The first and foremost is to avoid catastrophic error of computer vision systems in the real world. A strong causal understanding of a network before deployment can help with avoiding costly mistakes. Again, a good example is autonomous vehicle navigation: knowing that some naturalistic variations highly degrade the system would allow for patches to weaknesses, more strenuous real world testing prior to deployment or in the extreme to withhold deployment until the system is ready.

Another positive application is in fairness issues in networks from a counterfactual perspective, where a single attribute can be modified and network predictions can be observed post-modification. The field of counterfactual fairness [4, 1, 2, 6] primarily seeks to address this problem.

## Simulator and Assets

The simulator used to generate the NVD dataset is the MIT ThreeDWorld (TDW) simulator [3]. All of the objects contained in the NVD scenes are part of the full, or "non-free" TDW Model Library. These object models are licensed by the owners of TDW and are distributed for research purposes. The TDW code is public and distributed using a BSD 2-Clause "Simplified" License.