# OpenReview forum: "Finding Differences Between Transformers and ConvNets Using Counterfactual Simulation Testing"
_NeurIPS.cc/2022/Conference — NeurIPS 2022 Accept_

### Official Review · Reviewer_GX4W · 2022-07-06

**Rating:** 7
**Confidence:** 4
**Soundness:** 4 excellent
**Presentation:** 3 good
**Contribution:** 3 good

**Summary:**

The paper compares the robustness of Vision Transformers to Convolutional Networks under different transformations such as scaling, object rotation, camera changes, occlusion, and random patch deletion. The authors resort to a synthetic dataset to perform these transformations. Specifically, they use the MIT ThreeDWorld scene generator. The main finding is that ConvNext is often more robust than Swin – except for the occlusion task.

**Questions:**

These questions were also stated less explicit in the previous section:

- Do you plan to open source your NVD dataset and the corresponding code?
- How do the confidence intervals look for Figure 3-8?

**Limitations:**

The paper includes a limitation section, which I highly appreciate. The main limitations listed are the problem of how representative a synthetic dataset is and the difficulty of comparing different network architectures. It could also be mentioned that they only compare one specific training seed per network model, e.g., for each network architecture, only a single parameter set is used. However, I do not consider this a significant limitation as they included 5 different network architectures for Swin and ConvNext. Additionally, the authors might be excused as retraining them would be costly. Still, I wonder how much variation can be explained by the specific weight parameters.



**Strengths And Weaknesses:**

Overall this is a fine paper. The authors selected the two similar architectures ConvNex and Swin. The comparison using a synthetic photo-realistic dataset makes sense. I also liked that differently sized networks were compared. The paper is well written and original as it is the first study comparing Convolutional Networks to Vision Transformers on these transformations. Studying the robustness is also a significant problem. The approach to rely on a synthetic dataset gives the authors a high degree of control and therefore experimental validity. However, its results are a bit predictable when you know the "Appendix B. Robustness Evaluation" from the ConvNeXt paper. There, it is reported that ConvNeXt generalizes better to different datasets (ImageNet-A/R/C/Sketch) than Swin.



A few improvements are:

- Figures 3-8 lack confidence intervals; including them is necessary to judge how substantial the differences are. For example,  they might invalidate the conclusion from in Figure 6 that "ConvNext contain a higher proportion of conserved correct predictions". I would recommend generating them using bootstrap resampling.
- How large is the NVD dataset? 92 3D-models with 27 lighting conditions, thus 2484 images? Could you please include the number of samples in the paper?
- The conclusion needs to be extended. It reads more like an abstract than an informative summary of the findings and main takeaways. Some parts of the Discussion are included in the Related Work section, which goes against the more common structure to combine it with the conclusion.

My main criticism is the missing confidence intervals. Adding them might change the conclusion slightly, but I suspect that the overall story ("ConvNext models tend to be more robust than Swin models") still holds. Please add them as this would strengthen your analysis.

Additionally, it is unclear to me if the authors plan to make the NVD dataset accessible to the public? As they claim the dataset as a major contribution, I would expect this, but it is not stated explicitly in the paper.

Although, I generally liked this paper, I would like to vote for it only with a weak accept for now due to the missing confidence intervals. I would be willing to upgrade the score if my concern is address accordingly.

Minor Points:

- The font size of the figures is tiny and cannot be read when printed. Maybe, you could save the figures in the pgf format with the correct figure size (see https://jwalton.info/Matplotlib-latex-PGF/).
- Additionally, the y-axis of each Figure 3-8 should be fixed. For Figure 6, the y-axis changes from Tiny to Small, making it hard to compare them visually.

---

> ### Author Response · Authors · 2022-08-02
> **Response**
>
> We thank the reviewer for the positive comments and the thorough review. We are happy that they appreciated the problem we study, the quality of our work, the originality of the paper and the writing. We also thank them for their insightful comments and feedback.
>
> **Add confidence intervals to figures (bootstrap resampling)** \
> A strong suggestion. We re-plot all figures with error bars using bootstrap resampling (100 resamples). Error bars plotted are the standard deviation of all samples. We find almost identical curves to the original ones, with very small error bars for all experiments. We believe this is due to the large size of our simulated dataset. For example the occlusion experiment contains 92 main objects, 27 lighting environments and 3 occluder objects, which generates a total of ~7.4k object-occluder-lighting combinations for each occluder position (i.e. for each point in the graph). We observe that the conclusions of our work are conserved.
>
> **How large is NVD?** \
> We provided numbers for the amount of samples in the NVD dataset (272k total images) at the end of the Introduction and in the caption of Figure 1, but we did not re-iterate them in the Dataset section, which is important. We will add this and highlight it. Thank you.
>
> **Conclusion needs to be extended - it currently reads more like abstract. Make it informative summary of findings. Move discussion in related work to the conclusion.** \
> We have revamped the conclusion to follow this recommendation and have added the key findings of the work. We have also moved the Limitations section under the conclusion.
>
> **Will NVD be public?** \
> Yes, we will release NVD upon acceptance. We will also open-source the code used to generate the NVD dataset in order for other research teams to generate new variations or add object assets if they wish. We will highlight this in the introduction and method sections. Thank you for the remark.
>
> **Font size in figures is tiny** \
> Thank you for the suggestion. We will make them larger in the camera ready version. We saved the figures with high dpi using matplotlib, but we will use the suggested method for the final version of the work.
>
> **Figure y-axis** \
> Thank you for this suggestion. We have changed the figures in the main paper to have the same y-axis. This gives a lens into the differences between sizes of architectures, which we believe is very interesting. For example, we can clearly see increased performance and robustness as architecture size goes up. Also, importantly, when networks are trained on the ImageNet-22k dataset robustness in all categories goes up.
>
> **Try different training seeds for models** \
> This comment is on point. We had the same question early during our investigation. It is very expensive to retrain Swin and ConvNext models with limited resources. Even the tiny versions of the models can take >5 days depending on the hardware that is available. We were able to source ConvNext-Tiny and ConvNext-Base network trained using another codebase (timm codebase), with a different random seed and different hardware. We verified two things (1) ImageNet accuracies between the official checkpoints and the re-trained checkpoints differed only slightly (ConvNext-T:  81.9 vs reported acc 82.1 / ConvNext-B: 83.5 vs reported acc 83.8). (2) trends on NVD did not change.

---

> > ### Comment · Reviewer_GX4W · 2022-08-03
> > **Re: Rebuttal**
> >
> > Thank you for your clarifications regarding open sourcing NVD. I also appreciate the updated figures and conclusion.
> >
> > My suggestion regarding the confidence intervals was meant a bit different. I did not want to suggest re-training the models, but only to quantify the confidence given your finite test set. My concern was that the test set was relatively small, so that the confidence interval would be rather broad. However, given that you have 272k samples in total, this should not be a real concern. I would still suggest including them: you can use bootstrap resampling to construct them (https://acclab.github.io/bootstrap-confidence-intervals.html ).
> >
> > I agree that using different train seeds would be desirable but also highly computationally expensive. I appreciate that you cross-checked your results with the models from the timm's codebase.

---

> > > ### Author Response · Authors · 2022-08-03
> > > **Response**
> > >
> > > On confidence intervals, I believe we correctly understood the initial suggestion, we think that it is a good idea and that is what we have done (i.e. bootstrap resampling on the finite test set without re-training). The new figures are included in the revised paper. You might have to zoom in quite a bit since the error bars are very small for some of the figures. We also decreased the plot line width in order for the error bars to be more visible. We also agree that given the size of the dataset the variance is not a large concern but we think that it is a good idea to include the error bars.

---

> > > > ### Comment · Reviewer_GX4W · 2022-08-06
> > > > **Re: Confidence Intervals**
> > > >
> > > > Thank you for including confidence intervals. I did not zoomed into enough. Make sure you mention the confidence intervals shortly in the caption in a possible camera ready version.

---

> > > > > ### Author Response · Authors · 2022-08-06
> > > > > **Response**
> > > > >
> > > > > Absolutely, we will do so. Thank you again for your time and the thorough review.

---

### Official Review · Reviewer_j5VK · 2022-07-10

**Rating:** 5
**Confidence:** 4
**Soundness:** 3 good
**Presentation:** 3 good
**Contribution:** 3 good

**Summary:**

This paper presents a comparative study of ConvNext and Swin Transformer (with comparable model size and GMACs, and design and training techniques) on synthetic images with controlled scene variations. To avoid real-to-synthetic domain gap, the authors propose to measure (relative) accuracy drop (or robustness) on synthetic images with varying scene parameters, i.e. the so called counterfactual simulation testing and the proposed metric called proportion of correct conserved predictions. Therefore another contribution is the scene variation dataset generated with ThreeDWorld simulator, where the detailed description of what type of scene variations are provided. The comparison results show the differences to object viewpoint, scale and occlusions.

**Questions:**

1. The last column in Table 1 shows the top-5 accuracy when test on NVD. How about top-1?
2. Since all results on robustness metrics are based on top-5 predictions, and ConvNext is better than Swin Transformer on top-5, will most results on PCCP affected by this large difference or not? eg. Fig 4-7 almost all show ConvNext is better than SwinT with variations to those factors. How unstable the results would be if using Top-1?
3. Are the selected classes in NVD dataset representative in ImageNet?
4. Have you tried/what's your guess: will the results (table 1 and other Figs) transfer to real dataset with scene variations as well? (It would be much stronger if there is results on real validation data)
5. To clarify: for maximal occlusion (x=0), how much visible is the object of interest?
6. Can author put the concise main results/observations about the difference at the end of the abstract?

**Limitations:**

- related work (disclaimer: not related to myself) also looks at effect of scene parameter variations (in-distribution): Madan, Spandan, et al. "Small in-distribution changes in 3D perspective and lighting fool both CNNs and Transformers." arXiv preprint arXiv:2106.16198 (2021).
- please add a link to dataset generator and license in the main paper.

**Strengths And Weaknesses:**

Strengths
- Selection of ConvNext and Swin transformer is a fair choice for the comparative study than previous works, also tested SwinT v2 in the appendix.
- The proposed metrics largely avoids the real-to-synthetic gap.
- Careful choices of scene variations to occlusion, object pose, scale, camera poses, more comprehensive than previous studies. e.g. occlusion is better with occluder objects than random patch drop.

Weaknesses
- The performance gap on several variations (e.g. Fig 4, 6) are not significant enough to make a conclusion
- It would be better if further look into *learned features* similar to previous works on comparing ViT and ResNet to provide more insights on the difference, e.g. feature correlation, frequency, etc
- Dataset quality: although the dataset is called *Natural* variation object dataset, the synthetic images in Figure 1 do not look natural/realistic enough to me. Also scene background is relatively simple compared to real images. The number of object classes is rather small.
- Writing can be improved to be more concise and clearer.

---

> ### Author Response · Authors · 2022-08-02
> **Response 3/3**
>
> **Does the top5 performance advantage of convnext translate to PCCP?** \
> This is a good question. First, it is very hard to compare two different networks, especially in a domain different from the source domain. Our main motivation by designing the PCCP metric was to try to abstract from this exact performance advantage of ConvNext in this specific domain. We believe that we have been successful in some measure, given that ConvNext is better in some tasks than Swin, but Swin is better than ConvNext in others (occlusion). Finally, we would like to note that we compare the two most comparable architectures in the literature - with very close top1 and top5 accuracies in ImageNet.
>
> **Can authors put main conclusions at the end of the abstract?** \
> We have added a concise explanation of the main differences that we found between ConvNext and Swin networks in the abstract. Thank you for this suggestion.
>
> **For maximal occlusion x=0, how much of the main object is visible?** \
> Good question! The occluded percentage of the main object at x=0 is variable, between 80% and 100% - and we will add this to the dataset section of our work. In our simulated dataset we did not restrict objects to have the exact same volume, given that this would have reduced the realism overall (e.g. a computer mouse should be smaller than a laptop). We also decided against resizing the occluders mainly due to positioning issues between occluder/object/table. If the occluder were to be resized depending on the main object it would result in many instances of object clipping if the positioning schedule of the occluder was not independently designed for each main object. We sought the more general solution of having variable occluded percentages, with a static positioning schedule for the occluder. Finally, note that we have 92 main objects, 27 lighting environments and 3 occluder objects, we have a total of ~7.4k object-occluder-lighting combinations for each position of the occluder. We thank the reviewer for this comment and we will add all of these assumptions to the experiment section of our work.
>
> **Can authors put main conclusions at the end of the abstract?** \
> We have added a concise explanation of the main differences that we found between ConvNext and Swin networks in the abstract. Thank you for this suggestion.
>
> **Related work: add and discuss suggested.** \
> Thank you very much for bringing this work to our attention. We will gladly include this work in our related work section, with an appropriate discussion. We think this is very interesting work with some interesting conclusions. Our work differs in several ways (1) we study OOD generalization, while they purposefully study in-domain generalization with an unbiased training set (2) we both show weaknesses in current SotA networks, but the weaknesses are different: they show that learning using these networks inevitably causes the networks to underperform for certain variations, we show that current networks trained on vast amounts of real data are incredibly fragile with respect to all sorts of simple variations of data (3) they show that different types of networks fail in roughly the same ways, we also see that in our case but we make a special attempt to understand where different architectures fail in different ways by proposing NVD and our counterfactual metrics
>
> **Add link to dataset generator + license** \
> We will add a link to the TDW simulator in the camera ready version. We have also included our dataset generator code in the supplementary material and we will open-source this code upon acceptance. Finally, we will move the license information from the supplementary material to the main paper in the camera ready version (when more space is allowed for the main submission). We will also include the full license of the TDW simulator and assets in the README of the code.

---

> ### Author Response · Authors · 2022-08-02
> **Response 2/3**
>
> **Dataset quality: although the dataset is called Natural variation object dataset, the synthetic images in Figure 1 do not look natural/realistic enough to me. Also scene background is relatively simple compared to real images. The number of object classes is rather small.** \
> We call the dataset Naturalistic Variation Object Dataset, avoiding the word "natural" since we do agree that the dataset is not a natural dataset. Instead we opt for the word naturalistic since the variations that we propose (pose changes, occlusions, etc.) are found in the natural world and affect computer vision algorithms drastically. The word naturalistic is not supposed to reference the realism of the environment, but the plausibility of the variations included in the dataset. We do agree that there is a domain gap between real images and images in NVD - and we call out this fact several times in the work (l.68,118,122,etc.). In fact a large part of our contribution is the counterfactual metric that seeks to minimize the impact of the domain variation. We will call out this limitation further in the camera ready version. Further, we agree that scene variability is not as large as that seen in COCO for example - but we do this on purpose since we focus on a one-object classification task. We try to minimize the amount of first-order distractors in the scene while remaining realistic (instead of using a limited blank scene with only one object). We think the task is already challenging enough for the pre-trained networks, with low generalization accuracy overall and a more complicated setup would introduce much more noise to the signal. Finally, we point to our answer above (Are classes in NVD representative of ImageNet?) for a discussion on the number of object classes.
>
> **Why use top5 instead of top1? Please show top1 accuracies on table 1. What if you use top1 for the figures?** \
> We think that the top1 accuracy is an unreliable metric in our setting, and top5 is a much more appropriate metric to study. This is because our simulated images contain some first order distractors that are in the ImageNet label space. The primary distractor is the dining table where the objects are set. We decided to include these naturalistic distractors, in order to make the scene more realistic. An essentially blank scene without other objects would not make a convincing experiment. This means that analyzing top1 metrics is very noisy given that a network could output one of the distractors (e.g. the dining table) as its top1 prediction and this would not represent the overall power of the network which could be predicting both a distractor and the main object with high confidence. We have verified that this exact phenomenon occurs. Nevertheless, we have included the top1 accuracy on Table 1 and plotted figures for top1 accuracy that will be included in the supplementary material with this explanation on why we think these figures contain more noise than signal.
>
> **What is your guess/try to see if results in table 1 and figs are the same on real data?** \
> There are several considerations on why a real experiment is hard to control. The first consideration is scale. In our NVD dataset we have 272k images. Assuming a person has to manipulate an object or camera between each picture, and assuming the time spent between pictures is around 10 seconds, this would equate to around 755 labor hours without breaks. Assuming this is split into a group of 10 people, we would have 75.5 hours of labor per person at the low end. This is very hard to achieve. Alternatively, an automatic capture setup would have to be devised, which would probably be able to achieve this at a faster rate. A usual limitation of this type of approach is that the scene is not a natural environment, but instead a lab environment. Also, the conceptualization and realization costs are high. The next consideration is precision, it’s hard to move the camera or object in precise measurements. And some variations such as scale are not easily feasible. The last consideration is the consistency of some variations, such as controlling the lighting to be exactly the same for all images with the same lighting environment. This would require a room with no windows and artificial lighting that will be very different from ambient or natural lighting.
>
> Nevertheless, we believe that given the scale of our experiments, if we set them up using the same scale of real data, we would have very similar results and the same conclusions. Unfortunately, currently there are no real dataset with realistic variations that have the sufficient scale to make strong conclusions. We believe this is interesting future work. Finally, we would like to note that we do have an experiment on real data for patch occlusions in Figure 3, and that this experiment echoes the findings in our simulated occlusion experiment, giving us validation that there is transfer between domains.

---

> ### Author Response · Authors · 2022-08-02
> **Response 1/3**
>
> We thank the reviewer for the comments and the thorough review. We are happy that they found that we select good architectures for comparison, and that we have a careful selection of scene variations that are more comprehensive than related work. Finally, we thank the reviewer for their positive comments on our proposed metric and for the insightful feedback. We will incorporate all proposed changes.
>
> **The performance gap on several variations (e.g. Fig 4, 6) are not significant enough to make a conclusion** \
> We believe the variations that lead us to claim our main conclusions (e.g. Fig 4, 6) are significant enough. We follow R3's suggestion to include error bars using 100 bootstrap samples for all of our plots. We observe error bars in Fig 4,6 are very small and indicate that the performance gap is statistically significant. We don't see any change in our conclusions given the size of the uncertainty. We also highlight that the amount of images tested for these experiments is very high (88k for Fig 4 and 44k for Fig 6).
>
> **It would be better if further look into learned features similar to previous works on comparing ViT and ResNet to provide more insights on the difference, e.g. feature correlation, frequency, etc** \
> We agree that this would be a very interesting future work. Specifically, we would think an interesting avenue would be to zoom in on one specific experiment, let's say occlusion, and to study the variation of features. One key difficulty here, that is still an open problem, is to know which type of feature analysis makes sense and gives an interpretable lens into the inner workings of the networks. This problem is challenging enough that it spans different subfields of computer vision (interpretability, explainability, neuron level interpretability, causality, etc.). Another key difficulty is: how should features be compared across such different architectures? Our study abstracts from these two problems by directly looking at prediction results, which are a closer proxy to model generalization than model features. Finally, this research avenue does add a key difficulty in our situation, given that ConvNext generalizes better in the simulated setting. A method, in a similar vein to PCCP, would have to be invented in order to abstract from the better domain generalization of one architecture. We leave this interesting exploration for future work and we agree that it is a challenging but exciting problem. We have some initial thoughts on conducting this research, with planned explorations in the mapping of visual saliency.
>
> **Are classes in NVD representative of ImageNet?** \
> All objects included in our simulated dataset are included in the ImageNet label space. In this sense they are a strict subset of ImageNet classes. We were graciously given access to the full set of object assets by the ThreeDWorld owners for this work. We parsed the entire list of assets and found all objects that mapped to an ImageNet class. We then filtered objects that were not suitable due to extremely low recognition levels for both networks (e.g. modern iPods that did not exist when ImageNet was released) and anomalous objects with incorrect scale. We end up with 92 objects from 18 classes. It is important to note that this is small compared to the 1k classes in ImageNet and we will add more discussion about this in the camera ready version. We have done our best effort to include the maximum number of classes that were available to us, given the restricted amount of realistic assets that exist for this type of study. We have done a thorough online search for more object assets and we found no compatible asset package online that included (1) enough ImageNet class objects (2) that were realistic enough. We would also like to note that our work contains more ImageNet object classes, scene variations and lighting variations than related work in the same vein.
>
> Further, it is hard to know what would make a representative sample of classes from ImageNet. First, we work in an indoor environment with inanimate objects - which is a realistic scenario. This does not fully align with ImageNet, since ImageNet has a very large amount of animal images with many represented subspecies. We believe our scenario is in some sense more aligned with modern applications of computer vision that deal with objects in households (e.g. robotics, home assistants, etc.).
>
> Finally, thanks to R3's suggestion, we include error bars for all of our plots via bootstrap resampling. This shows that the variance of the results is very small, even when some of the classes we use are under/oversampled in the bootstrap sample.

---

### Official Review · Reviewer_MNdi · 2022-07-11

**Rating:** 6
**Confidence:** 3
**Soundness:** 4 excellent
**Presentation:** 3 good
**Contribution:** 3 good

**Summary:**

This paper studies the robustness of different Imagenet pre-trained classifiers to changes in object scale, object pose, scene lighting, and 3D occlusion. The authors generate a large dataset of 272k synthetic images ("NVD") to do so, which is the central contribution of the paper. The 2 architectures primarily under study are ConvNext architectures and Swin ViT architectures. The authors first show that ConvNext models generalize significantly better to NVD than Swin Transformer models. The authors then show that -- when accounting for the different affects of the real-to-synthetic domain gap -- Swin Transformers are more robust to occlusion, but ConvNext models are generally more robust to variation in object scale, changes in viewpoint pitch and changes in viewpoint yaw.

**Questions:**

- Figures 3-8 - to some extent, I feel the authors should keep the same y-axis limits across the subplots in each of these figures. Or at least provide an additional plot where all curves are all on the same plot? It would be good to understand better how differences between the two architectures scale with model size (e.g. are large CNNs and large ViTs more similar to each other than small CNNs and small ViTs?), which is hard to parse from the current plots.
- NVD - the authors should make it clear whether or not they plan to release to the public this dataset along with their code for computing metrics.

**Limitations:**

Yes, for instance the authors discussed that they primarily study generalization differences between ConvNext and Swin architectures which is not necessarily indicative of generalization differences between general CNNs and ViTs.

**Strengths And Weaknesses:**

I found this paper enjoyable and interesting to read. In my opinion, some of the results are very likely to be compelling to the ML community, such as the result that ConvNext architectures are much more robust to random patch drop and synthetic object occlusion than the older CNN architectures tested in prior work (Figures 3-4). The result that ConvNext architectures generalize significantly better than Swin architectures to the synthetic data is also an interesting finding, and counters prior work suggesting ViTs are better for OOD generalization (Table 1).

I have a few suggestions that I think would further improve the impact of the paper:
- In lines 124-149, the authors make the argument that their PCCP metric marginalizes out the differing effects of the real-to-synthetic domain gap in order to solely investigate performance under changes to object scale and viewpoint etc. I think this point is very important for the paper and some justification is necessary to support this claim. I'm not entirely convinced that this metric does enough to account for that gap - isn't it still feasible that how sensitive a model is to changes in viewpoint on the synthetic data does not correlate with the model's sensitivity on the real data? I'm wondering if the authors could maybe post-process the extent to which an object is occluded (maybe with a segmentation model?) in the Imagenet validation set and verify that the "performance vs. occlusion" curves on these natural  images matches the curves in Figure 4.
- To further improve the impact of the paper, I recommend that the authors additionally test how the curves in Figures 4-8 shift when models are fine-tuned on the synthetic data. In practice, if one finds a subclass of images where the model is performing poorly, the next step is to train on more images of that type. It's therefore very practically useful to understand if the difference in performance between the two architectures vanishes or widens when both architectures are fine-tuned for a small number of steps.

---

> ### Author Response · Authors · 2022-08-02
> **Response 2/2**
>
> **NVD: will it be released?** \
> Yes, we will release NVD upon acceptance. We will also open-source the code used to generate the NVD dataset in order for other research teams to generate new variations or add object assets if they wish. We will highlight this in the introduction and dataset sections. Thank you for the remark.
>
> **Authors discussed that they primarily study generalization differences between ConvNext and Swin architectures which is not necessarily indicative of generalization differences between general CNNs and ViTs.** \
> We agree with this statement of our limitation and point to it in our Limitations section. We will add a further specific callout in this same section in the camera ready version saying that the conclusions only apply for the architectures that we study in the work (ConvNext, Swin, Swin-v2).

---

> > ### Comment · Reviewer_MNdi · 2022-08-07
> > **Thank you to the authors**
> >
> > I appreciate the authors' response to my questions and comments and their speed in updating their submission. I think this is a well-written study on differences between ViTs and CNNs. I currently feel though that without an experiment that grounds these results on real-world data, I have some uncertainty about how well these trends transfer to general image classification. I'll maintain my score.

---

> ### Author Response · Authors · 2022-08-02
> **Response 1/2**
>
> We thank the reviewer for their comments. We are happy that the reviewer found our work enjoyable and interesting to read. We are encouraged that they found that some of our results are likely to be very compelling for the ML community. Finally, we thank them for their valuable suggestions.
>
> **Suggestion: Finetuning networks on simulated images.** \
> We believe this is a very promising study, although these are some considerations at hand: (1) our work tries to study out-of-the-box generalization performance, which is why we do not finetune and test on objects the networks have never seen before (2) when finetuning it is hard to determine how long the network should be finetuned such that it doesn't overfit the object+lighting+pose. (3) should the network be trained on some objects per class and try to generalize to others? In this case NVD has a limited amount of objects and this could prove tricky.
>
> This being said, we run a version of this experiment and include it in the revised supplementary material. Specifically, we finetune all Swin and Convnext networks on a dataset composed of all the objects in NVD, under bright lighting and in a canonical view. We use 30 epochs, with the same learning rates across architectures (5e-5 for Tiny, Small / 2e-5 for Base, Large). We then run the object rotation simulated experiment on these networks. We find very similar conclusions to the original, non-finetuned experiment presented in Fig. 8: ConvNext networks are more robust to object pose changes on average than Swin networks. There is one peculiar difference. It seems that smaller ConvNext networks have overfitted slightly to a canonical view of the object, with harsh drop rates for a specific rare pose (around 180 degrees, when the object is fully turned around). This is an interesting phenomenon that is worth investigating further and we thank the reviewer for their suggestion. For more details please find the experiment at the end of the supplementary material.
>
> We believe this experiment deserves special care to design and we think future work can more thoroughly study this scenario. Nevertheless, we can run all the simulated experiments using this setup and include them in the supplementary material for the camera-ready with an analysis.
>
> **It is possible for PCCP to not marginalize domain shift effects. Suggestion: run occlusion experiment on ImageNet val.** \
> We thank the reviewer for their comment. It is true that there exists no metric that would be able to perfectly marginalize domain shift effects, nevertheless we believe it is an important problem to tackle and we propose an attempt that we believe is much better than naive comparison of accuracies. We also have evidence (thanks to the reviewer’s suggestion of finetuning, addressed above) that we retrieve the same conclusions even when the networks have been finetuned on simulated objects. Finally, we believe the scale of our dataset mitigates this problem given that there is lower likelihood that this would happen across 2.5k object/lighting combinations.
>
> We believe such an occlusion experiment on ImageNet would be valuable, but there are some considerations: (1) If we use a segmentation network in order to occlude parts of an object in the ImageNet dataset we have to subject ourselves to the fail-rate of that specific segmentation network, which might bias results (2) Further, the segmentation network might have failures that are correlated to the failures of ImageNet-trained classifiers, thus filtering out important hard test cases (3) We have asked ourselves this same question, and have explored datasets to study occlusion on real images. To our surprise we found no dataset that was suited for this task, and most real studies for occlusion on data are handled in similar ways to the patch occlusion experiment. (4) In our work, we do the next best thing which is an occlusion experiment using patch occlusion on ImageNet, since this method does not depend on an auxiliary network - and we find corroborative evidence for our simulated occlusion experiment (i.e. Swin is more robust).
>
> **Figs: use the same y-axis. Maybe a new plot for that?** \
> Thank you for this strong suggestion. We have changed the figures in the main paper to have the same y-axis. This gives a lens into the differences between sizes of architectures, which we believe is very interesting. For example, we can clearly see increased performance and robustness as architecture size goes up. Also, importantly, when networks are trained on the ImageNet-22k dataset robustness in all categories goes up.

---

### Public Comment · ~Nataniel_Ruiz1 · 2022-11-21
**Wrong project page URL**

Project page URL in current version is erroneous, please use https://counterfactualsimulation.github.io instead

---

### Meta-Review · Area_Chair_XAD4 · 2022-08-25

**Recommendation:** Accept
**Confidence:** Certain

**Metareview:**

After the rebuttal and discussion all reviewers are positive, and recommend acceptance. The AC agrees with this recommendation.

**Award:**

No

---

### Decision · Program_Chairs · 2022-09-14

Accept